# Dynamic tracking and identification of tissue-specific secretory proteins in the circulation of live mice

Kwang-eun Kim [1,5], Isaac Park [2,5], Jeesoo Kim [3,4], Myeong-Gyun Kang [2], Won Gun Choi [1], Hyemi Shin [1], Jong-Seo Kim [3,4✉], Hyun-Woo Rhee [2,4✉] & Jae Myoung Suh [1✉]

Secretory proteins are an essential component of interorgan communication networks that regulate animal physiology. Current approaches for identifying secretory proteins from specific cell and tissue types are largely limited to in vitro or ex vivo models which often fail to recapitulate in vivo biology. As such, there is mounting interest in developing in vivo analytical tools that can provide accurate information on the origin, identity, and spatiotemporal dynamics of secretory proteins. Here, we describe iSLET (in situ Secretory protein Labeling via ER-anchored TurboID) which selectively labels proteins that transit through the classical secretory pathway via catalytic actions of Sec61b-TurboID, a proximity labeling enzyme anchored in the ER lumen. To validate iSLET in a whole-body system, we express iSLET in the mouse liver and demonstrate efficient labeling of liver secretory proteins which could be tracked and identified within circulating blood plasma. Furthermore, proteomic analysis of the labeled liver secretome enriched from liver iSLET mouse plasma is highly consistent with previous reports of liver secretory protein profiles. Taken together, iSLET is a versatile and powerful tool for studying spatiotemporal dynamics of secretory proteins, a valuable class of biomarkers and therapeutic targets.

[1] Graduate School of Medical Science and Engineering, KAIST, Daejeon, Republic of Korea. [2] Department of Chemistry, Seoul National University, Seoul, Republic of Korea. [3] Center for RNA Research, Institute for Basic Science, Seoul, Republic of Korea. [4] School of Biological Sciences, Seoul National University, Seoul, Republic of Korea. [5] These authors contributed equally: Kwang-eun Kim, Isaac Park. ✉email: jongseokim@snu.ac.kr; rheehw@snu.ac.kr; jmsuh@kaist.ac.kr

Secretory proteins released into the blood circulation play essential roles in physiological systems and are core mediators of interorgan communication[1]. To investigate this critical class of proteins, previous studies analyze conditioned media from in vitro or ex vivo culture models to identify cell type-specific secretory proteins, but these models often fail to fully recapitulate the intricacies of multi-organ systems and thus do not sufficiently reflect in vivo realities[2]. In other approaches, bioinformatic tools such as QENIE (Quantitative Endocrine Network Interaction Estimation) have been developed[3], however, in silico predictions of endocrine protein factors still require many additional layers of experimental validation. These limitations provide compelling motivation to develop in vivo techniques that can identify and resolve characteristics of tissue-specific secretory proteins along time and space dimensions.

To address this gap in current methodology, we sought to utilize recently developed proximity-labeling enzymes such as engineered biotin ligase (BioID)[4] or ascorbate peroxidase (APEX)[5]. When provided appropriate substrates, these enzymes generate reactive biotin species, leading to in situ biotinylation of proximal proteins on lysine or tyrosine residues, respectively. Thereafter, the biotinylated proteins are readily enriched through streptavidin affinity purification and can be identified through mass spectrometry. Recently, TurboID, a newly engineered biotin ligase, was developed to overcome the low biotinylation efficiency of BioID in subcellular compartments[6]. TurboID exhibits efficient proximity labeling activity across a broad range of subcellular compartments and possesses >100-fold improvement in labeling efficiency compared to that of BioID in cultured human cells[6]. Here, we introduce a method to profile tissue-specific secretory proteins in live mice by in situ proximity labeling of ER lumen proteins through the catalytic actions of an ER-anchored TurboID.

## Results

**Proximity labeling of secretory pathway proteins using ER-anchored TurboID.** To engineer a TurboID-based tool for the selective labeling of secretory proteins as they transit through the ER lumen, we first tested the functionality of two ER lumen-targeted TurboIDs, an ER lumen-localized TurboID (TurboID-V5-KDEL) and an ER membrane-anchored TurboID (Sec61b-V5-TurboID), in cultured cells. KDEL is a four amino acid sequence that acts as an ER retention signal when fused to the C-terminus of a target protein. On the other hand, Sec61b (Protein transport protein Sec61 subunit beta) is a single-pass ER transmembrane protein with its N-terminus and C-terminus exposed to the cytosol and ER lumen, respectively. Therefore, a fusion protein containing TurboID at the C-terminus of Sec61b is predicted to place the TurboID domain within the ER lumen. We transfected either TurboID-KDEL or Sec61b-TurboID expression constructs, both of which also express a V5 epitope tag, to cultured mammalian cells and analyzed biotinylated proteins in cell lysates and culture supernatant (Fig. 1a and Supplementary Fig. 1a). Immunofluorescence analysis of transfected cells with anti-V5 antibody and fluorescence-conjugated streptavidin confirmed expected patterns of ER localization for both TurboID-KDEL and Sec61b-TurboID along with their biotinylated targets (Fig. 1b). These results are consistent with previous ER localization studies for APEX2-KDEL and Sec61b-APEX2[7] and are in contrast with the cytosolic localization pattern of TurboID fused to a nuclear export signal (NES), TurboID-V5-NES, (Fig. 1b and Supplementary Fig. 1a). We further performed peroxidase-assisted EM imaging[8] to confirm the topology of Sec61b-TurboID at a higher resolution. HEK293 cells expressing APEX2 fused to the C-terminus of Sec61b-TurboID (N-Sec61b-TurboID-APEX2-C) (Supplementary Fig. 1a) revealed peroxidase staining patterns confined to the ER lumen (Supplementary Fig. 1b), similar to previous EM studies[8,9].

Streptavidin-HRP detection of biotinylated proteins in control cell lysates revealed the presence of several endogenous biotinylated carboxylases[10] which were not detected in the culture supernatant, indicating that these carboxylases are not secreted (Fig. 1c). In contrast, to control cells, a broad array of biotinylated proteins was detected in both cell lysates and culture supernatant of cells expressing TurboID-KDEL and Sec61b-TurboID in a biotin treatment-dependent manner (Fig. 1c). The secretory proteins labeled by Sec61b-TurboID were mostly newly synthesized proteins as cycloheximide treatment led to a dramatic reduction in biotinylated proteins in the culture supernatant. (Supplementary Fig. 1c).

Somewhat unexpectedly, TurboID-KDEL was readily detectable in the culture supernatant of TurboID-KDEL expressing cells treated with biotin (Fig. 1c), indicating incomplete retention of TurboID-KDEL in the ER lumen and subsequent release of TurboID-KDEL into culture supernatant (Fig. 1c). On the other hand, the ER-anchored Sec61b-TurboID was undetectable in the culture supernatant (Fig. 1c and Supplementary Fig. 2). These data demonstrate the effective retention of Sec61b-TurboID, but not TurboID-KDEL, in the ER compartment through ER membrane-tethering action of the Sec61b transmembrane domain. We also confirmed that Sec61b-TurboID robustly labeled secretory proteins without self-secretion in the HepG2 human liver cell line, whereas TurboID-KDEL was again found to be secreted into the culture supernatant (Supplementary Fig. 3).

Notably, the pattern of biotinylated proteins generated by Sec61b-TurboID in the culture supernatant was clearly different from that of whole-cell lysate, which is expected as ER-resident proteins and secretory proteins differ in composition (Fig. 1d). To further confirm the secretory pathway origin of Sec61b-TurboID biotinylated proteins, we treated HepG2 cells expressing Sec61b-TurboID with Brefeldin A (BFA), an inhibitor of ER to Golgi protein transport, and observed a uniform reduction in the amount of biotinylated proteins detected in the culture supernatant (Fig. 1e). Taken together, these data indicate that catalytically active Sec61b-TurboID is expressed and faithfully retained in the ER lumen, a necessary property for in vivo applications that require efficient and selective labeling of tissue-specific secretory proteins.

Labeling kinetics determined by biotin treatment time course studies indicate that Sec61b-TurboID efficiently labels secretory proteins in HepG2 cells by 10 min with increased labeling up to 4 h (Fig. 1f). Conversely, biotin washout time course studies indicate that Sec61b-TurboID labeled secretory proteins are largely sustained for 8 h (Fig. 1g and 1h). Therefore, Sec61b-TurboID can efficiently label classical secretory proteins in a biotin-dependent manner indicating compatibility with kinetic studies such as classical pulse-chase labeling analyses.

**Proteomic validation of secretory protein labeling using ER-anchored TurboID.** Next, we compared cytosolic labeling (TurboID-NES) (Supplementary Fig. 1a) and ER luminal labeling (Sec61b-TurboID) to confirm that our ER luminal labeling approach effectively differentiates canonical secreted proteins from cytoplasmic proteins. We performed proteomic analysis of biotinylated proteins enriched from the culture supernatant of HEK293 cells stably expressing either TurboID-NES or Sec61b-TurboID via liquid chromatography and tandem mass spectrometry (LC-MS/MS). Here, we utilized a previously optimized mass spectrometric analysis strategy, Spot-BioID[11,12], which

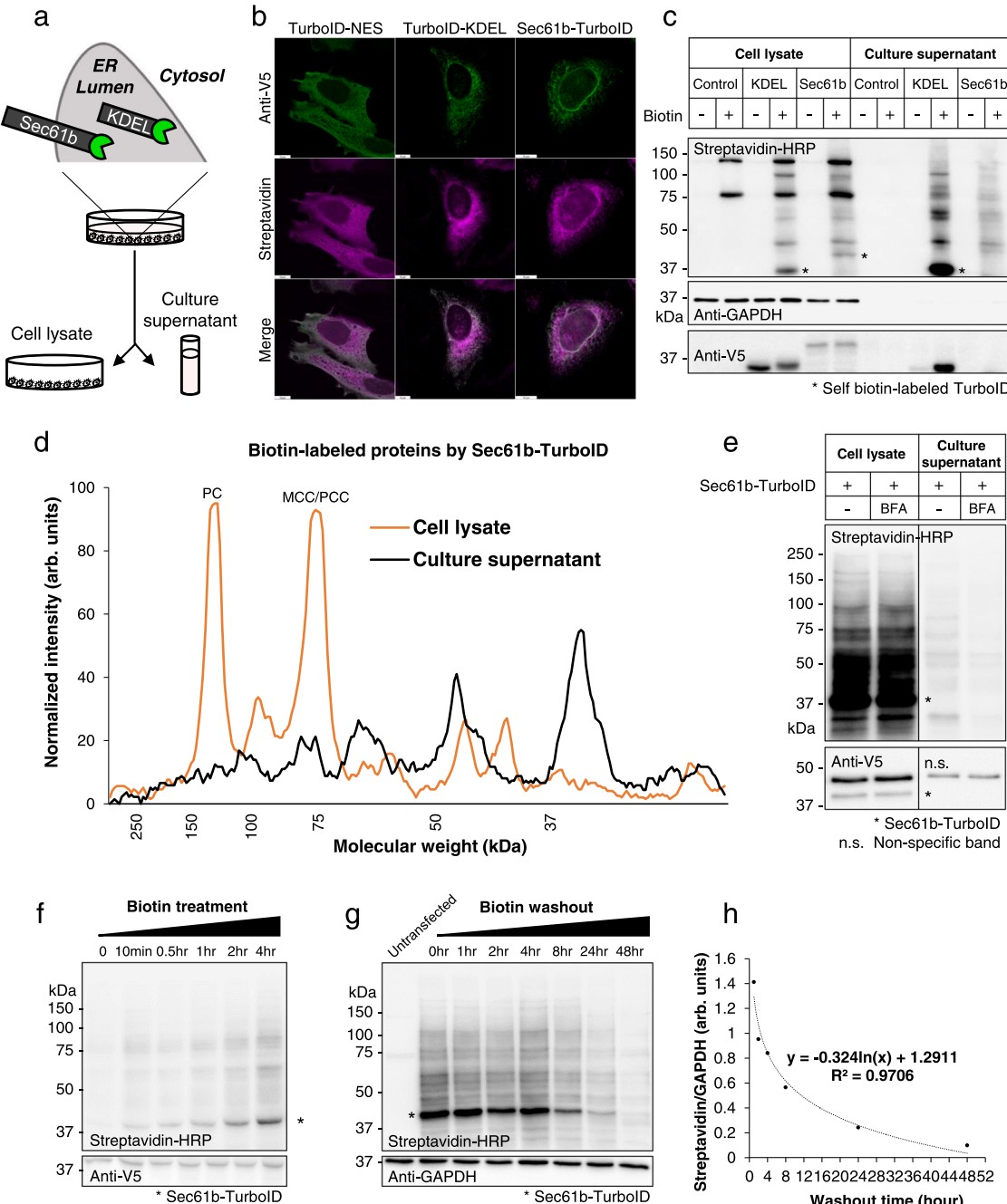

**Fig. 1 Proximity labeling of secretory pathway proteins using ER-anchored TurboID. a** Schematic illustration for secretory protein labeling by ER-localized TurboID (TurboID-KDEL) or ER-anchored TurboID (Sec61b-TurboID). **b** Immunofluorescence localization of TurboID (Anti-V5) and biotinylated proteins (Streptavidin-Alexa) in HeLa cells transfected with TurboID-V5-NES or TurboID-V5-KDEL or Sec61b-V5-TurboID expression plasmids. Scale bars, 10 μm. **c** Western blots for biotinylated proteins (Streptavidin-HRP) and TurboID (Anti-V5) in cell lysates or culture supernatant of NIH-3T3 cells transfected with GFP (Control), TurboID-V5-KDEL (KDEL), or Sec61b-V5-TurboID (Sec61b) expression plasmids. Anti-GAPDH as a loading control. Asterisk indicates self-biotinylated TurboID-KDEL or Sec61b-TurboID. **d** Line-scan analysis of biotinylated proteins in cell lysate (orange) or culture supernatant (black) from NIH-3T3 cells transfected with Sec61b-V5-TurboID expression plasmids and treated with biotin. PC, Pyruvate carboxylase; MCC/PCC, Methylcrotonyl-CoA Carboxylase/ Propionyl-CoA carboxylase. **e** Effect of Brefeldin A (BFA) on biotinylated protein secretion in HepG2 cells transfected with Sec61b-V5-TurboID expression plasmids. **f** Time course blot for biotinylated-labeled proteins (Streptavidin-HRP) in cell lysates of HepG2 cells transfected with Sec61b-V5-TurboID expression plasmids. **g** Time course blot for biotinylated protein (Streptavidin-HRP) turnover in cell lysates of HepG2 cells transfected with Sec61b-V5-TurboID expression plasmids following biotin washout. **h** Quantitation and plotting of the time course blot for biotinylated protein turnover shown in **g**. Asterisk indicates Sec61b-TurboID. Source data for **c**–**h** are provided as a Source Data file. These experiments were repeated as biological triplicates with similar results.

provides definitive evidence of biotinylation on the peptides by direct identification of the biotinylated lysine residue.

We detected 184 and 504 biotinylated proteins in the culture supernatant of TurboID-NES expressing cells and Sec61b-

TurboID expressing cells, respectively (Fig. 2a and b). Gene Ontology (GO) analysis showed that cytosol and cytoplasmic proteins were enriched in the culture supernatant of TurboID-NES expressing cells (Fig. 2c) whereas extracellular proteins were

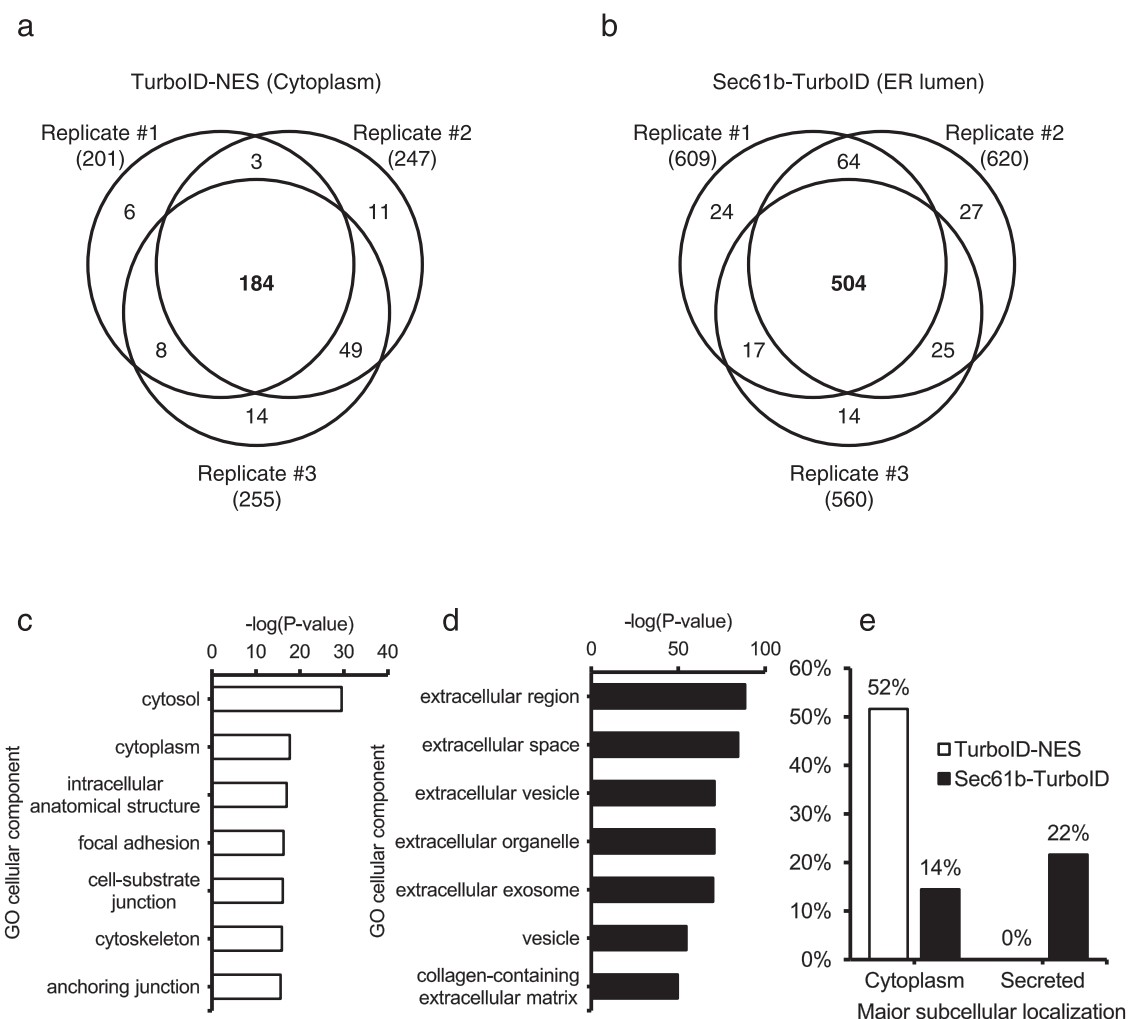

**Fig. 2 Proteomic validation of secretory protein labeling using ER-anchored TurboID. a**, **b** Venn diagrams depicting biotinylated proteins with more than two biotinylated spectral counts for each biological replicate in the culture supernatant of HEK293 cells expressing TurboID-NES (**a**) and Sec61b-TurboID (**b**), respectively. **c** Gene Ontology (GO) analysis of detected proteins in the culture supernatant of HEK293 cells expressing TurboID-NES. **d** Gene Ontology (GO) analysis of detected proteins in the culture supernatant of HEK293 cells expressing Sec61b-TurboID. **e** Fraction of cytoplasmic and secreted proteins in the culture supernatant of HEK293 cells expressing TurboID-NES or Sec61b-TurboID.

enriched in the culture supernatant of Sec61b-TurboID expressing cells (Fig. 2d). We also analyzed the annotated subcellular localization of detected proteins in each group and, as expected, the fraction of secreted proteins were highly enriched in Sec61b-TurboID expressing cells (Fig. 2e).

In addition, we performed a traditional secretome analysis of cell culture supernatant to compare coverage and specificity of ER luminal labeling (Supplementary Fig. 4a). Culture supernatant proteins from HEK293 cells were collected for traditional secretome analysis. Traditional secretome overlapped with biotinylated proteome data from cytoplasmic or ER luminal labeling (Supplementary Fig. 4b) and the fraction of cytoplasmic proteins and secreted proteins were 29% and 28% in the traditional secretome analysis (Supplementary Fig. 4c). These results demonstrate that Sec61b-TurboID labels canonical secretory proteins with similar coverage and specificity as the traditional secretome analysis in vitro.

**Identification of liver-specific secretory proteins in the circulation of liver *i*SLET mice.** Next, we applied our method, named *i*SLET, in situ Secretory Protein Labeling via ER-anchored TurboID, in live mice to demonstrate its in vivo functionality.

Sec61b-TurboID adenovirus was delivered to mice via tail vein injection to establish a liver *i*SLET mouse model and biotin was administered to these mice to induce labeling of liver secretory proteins (Fig. 3a). Because TurboID has faster labeling kinetics than BioID, we administered 24 mg/kg biotin to mice for 3 days, as compared to 5 to 7 days of biotin administration used for previous BioID labeling studies in the postnatal brain[13,14]. Four days after Sec61b-TurboID adenovirus delivery, we confirmed that Sec61b-TurboID (anti-V5) was expressed in the liver, and those liver proteins were efficiently biotinylated (Fig. 3b and Supplementary Fig. 5a). Liver tissues examined by histological analysis did not reveal any obvious adverse effects due to adenoviral overexpression of Sec61b-TurboID and biotin administration (Supplementary Fig. 5b). ER stress or apoptosis markers in liver tissue were also not changed by liver *i*SLET expression and biotin administration. (Supplementary Fig. 6).

Consistent with results obtained from the culture supernatant of Sec61b-TurboID-expressing cell lines, endogenous biotinylated proteins were not detected in plasma samples from liver *i*SLET mice (Fig. 3c). Thus, we could unambiguously detect Sec61b-TurboID-dependent biotinylated liver secretory proteins in the plasma from liver *i*SLET mice without any background (Fig. 3c and d). Interestingly, the pattern of biotinylated proteins secreted

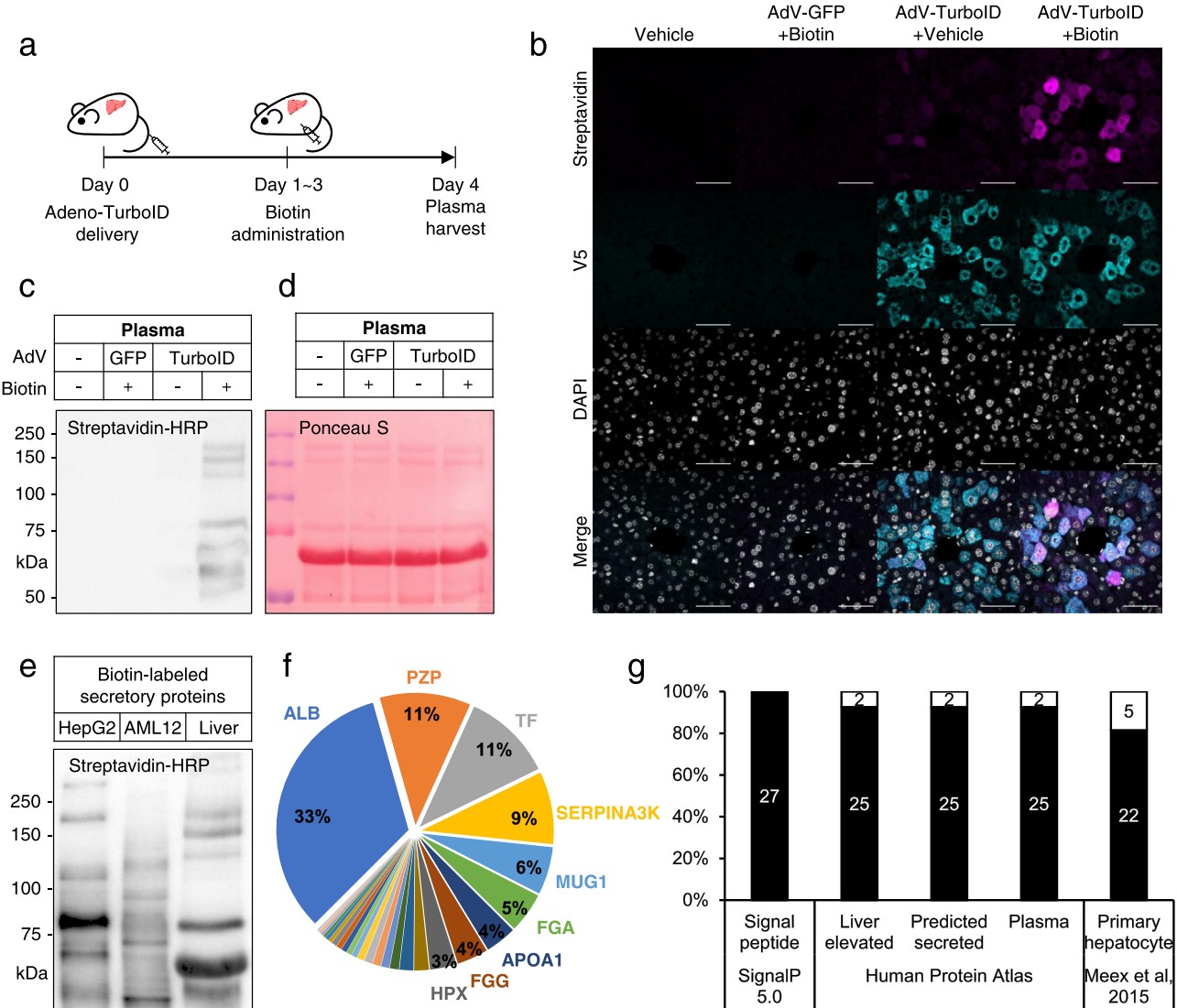

**Fig. 3 Identification of liver-specific secretory proteins in plasma from liver *i*SLET mice. a** Experimental scheme for adenoviral expression of Sec61b-V5-TurboID and biotin labeling in mouse liver tissue. **b** Immunofluorescence imaging of TurboID (Anti-V5) and biotinylated proteins (Streptavidin-Alexa) in liver tissue sections from GFP or Sec61b-V5-TurboID adenovirus (AdV) transduced mice. Scale bars, 50 μm. These experiments were repeated as biological triplicates with similar results. **c, d** Streptavidin-HRP detection of biotinylated proteins (**c**) and Ponceau S staining of proteins (**d**) in blood plasma from GFP or Sec61b-V5-TurboID adenovirus (AdV) transduced mice. These experiments were repeated as biological triplicates with similar results. **e** Biotinylated secretory protein profiles generated by Sec61b-V5-TurboID adenovirus in supernatants of hepatocyte cell lines, HepG2 and AML12, and plasma of liver *i*SLET mice. These experiments were repeated as biological triplicates with similar results. **f** Relative abundance of biotinylated secretory proteins detected in the plasma of liver *i*SLET mice in terms of biotinylated spectral counts. ALB, Serum albumin; PZP, Pregnancy zone protein; TF, Serotransferrin; SERPINA3K, Serine protease inhibitor A3k; MUG1, Murinoglobulin-1; FGA, Fibrinogen alpha chain; APOA1, Apolipoprotein A-I; FGG, Fibrinogen gamma chain; HPX, Hemopexin. **g** Specificity analysis for biotinylated proteins with SignalP 5.0, Human Protein Atlas (HPA), and literature.

from the liver in vivo was clearly distinct from that of hepatocyte cell lines, human HepG2, and mouse AML12 (Fig. 3e). These data confirm the in vivo functionality of Sec61b-TurboID as demonstrated by the detection of biotinylated secretory protein species released into the circulation of liver *i*SLET mice.

We next performed proteomic analysis of biotinylated proteins enriched from liver *i*SLET mice plasma by LC-MS/MS. Initial data from LC-MS/MS analysis of liver *i*SLET mice plasma identified nearly 190 proteins and among them, ~50 proteins were confirmed to be biotinylated, which were thereafter narrowed down to 27 proteins after applying more stringent criteria, i.e., more than three biotinylated spectral counts per protein (Supplementary Data 1 and Supplementary Data 2). On the other hand, we did not detect any biotinylated proteins in

control mice plasma which is consistent with the absence of bands in streptavidin-HRP blots in control plasma (Fig. 3c). Representative MS/MS spectra of the biotinylated peptides show the accurate identification of biotinylated residues (Supplementary Fig. 7 and Supplementary Data 2). Signal peptide analysis for biotinylated proteins with SignalP 5.0[15] revealed that all of the detected proteins contain signal peptides required for cotranslational transport to the ER lumen which indicates selective labeling of secretory proteins by *i*SLET (Fig. 3g).

As expected, serum albumin (ALB) was the most abundant biotinylated protein detected from liver *i*SLET mice plasma samples (Fig. 3f). Interestingly, the second most abundant protein was pregnancy zone protein (PZP, Q61838) (Fig. 3f), which is also annotated under the alias alpha-2-macroglobulin (A2M,

Q6GQT1) in the UniProt database. However, *Pzp* and *A2m* are independent genes in the mouse genome[16], and the identified peptides in our analysis were a precise match to the sequence of PZP but not A2M (Supplementary Fig. 7).

We found that 93% of the proteins identified in liver *i*SLET mice plasma are annotated as liver-enriched and predicted as secreted plasma proteins in the Human Protein Atlas database (Fig. 3g). We next compared the secretory protein profiles from liver *i*SLET mice plasma with ex vivo secretome studies using primary hepatocytes[17]. While a considerable fraction (81%) of proteins were common in both (Fig. 3g), fibrinogen gamma chain (FGA), complement component C8 alpha chain (C8A), histidine-rich glycoprotein (HRG), inter alpha-trypsin inhibitor, heavy chain 4 (ITIH4) and serine protease inhibitor A3M (SERPI-NA3M) were only detected in the plasma of liver *i*SLET mice (Supplementary Data 1). Taken together, our results indicate that the liver-specific secretory protein profiles obtained from liver *i*SLET mice are conserved in humans and more accurately reflect in vivo physiology compared to conventional ex vivo secretome analyses.

**Application of *i*SLET to pathophysiological mouse models**. We next sought to apply *i*SLET to characterize secreted proteomes associated with in vivo pathophysiology in which endocrine signals play an important role such as insulin resistance. S961 is an insulin receptor antagonist that induces systemic insulin resistance[18]. We administered S961 and biotin for 8 consecutive days to liver *i*SLET mice generated by adenoviral delivery of the Sec61-TurboID transgene (Fig. 4a). S691 administration to mice dramatically increased blood glucose confirming the insulin resistance state (Fig. 4b). LC-MS/MS analysis of plasma proteome from vehicle (PBS) or S961 group resulted in the identification of 25 and 45 biotinylated protein species, respectively (Fig. 4c and Supplementary Fig. 8). Notably, 20 of the identified proteins were exclusively found in the S961 administered insulin-resistant group. Among these proteins, many have been reported to play a role in the development of insulin resistance (Fig. 4d).

Alpha-2-HS-glycoprotein (AHSG), also known as Fetuin-A, is elevated in the serum of obese diabetic human subjects and induces insulin resistance[19]. Inter-alpha-trypsin inhibitor heavy chain H1 (ITIH1) is increased in human subjects with impaired glucose tolerance or diabetes and antibody neutralization of ITIH1 ameliorates systemic insulin resistance in mice[20]. Afamin (AFM) was found to have a significant association with insulin resistance, prevalence, and incidence of type 2 diabetes in a study from a pooled analysis of >20,000 individuals[21]. Beta-2-glycoprotein 1 (APOH), also known as apolipoprotein H is increased in the plasma of type 2 diabetic patients with metabolic syndrome[22]. For orthogonal validation of S961-induced plasma proteins identified in liver *i*SLET mice, we measured plasma concentrations of AHSG (Fetuin-A) and confirmed that AHSG was significantly increased in the mice treated with S961 (Fig. 4e). These results demonstrate that *i*SLET technology can be successfully applied to animal disease models for the discovery of tissue-specific secreted proteins with potential value as therapeutic targets or biomarkers.

The *i*SLET is an application of proximity labeling that enables dynamic tracking of tissue-specific secretory proteins in the circulation of live mice. Liver *i*SLET mice may be utilized to deepen our understanding of liver endocrine signaling by investigating secretory protein profiles under various physiological or disease conditions. Another valuable feature of *i*SLET technology is that it can be applied to longitudinal secretome profiling studies by drawing blood samples, which contain labeled secretome, at multiple time points from the same individual. Pre-immunodepletion of abundant plasma proteins such as ALB and PZP can further enhance the coverage of secretory protein profiles identified from *i*SLET studies. Further investigations to optimize the route, duration, and dosage of biotin administration may allow for improved labeling efficiency and customization of *i*SLET applications.

While our manuscript was under review, three similar papers were recently published. Wei et al.[23] used AAV-TurboID viruses for cell type-selective secretome profiling in mouse plasma secreted by hepatocyte, myocyte, pericyte, and myeloid cells. Liu et al.[24] generated ER-BioID mice and characterized muscle-derived serum protein changes with exercise training. Droujinine et al.[25] showed protein trafficking by tissue-specific labeling in *Drosophila* using BirA*G3 and identified serum proteins secreted from mouse teratomas. All these studies used the KDEL signal peptide to position TurboID, BioID, or BirA*G3 to the ER lumen whereas our study utilizes the membrane-anchored Sec6lb-TurboID which is not subject to secretion. Also, we enhance the specificity of the proteomic analysis through direct identification of biotinylated peptides by Spot-BioID[11,12] instead of conventional protein mass spectrometry, which relies on quantitative enrichment analysis for the identification of biotinylated proteins.

Taken together, *i*SLET is a versatile and adaptable in vivo approach to profile tissue-specific secretory proteins as *i*SLET expression in a tissue-of-interest can be achieved using a variety of existing conditional gene expression strategies[26]. We anticipate *i*SLET will be a valuable experimental tool for the identification of tissue-specific endocrine protein factors and the deconvolution of complex interorgan communication networks.

## Methods

**Animals**. All animal experiments were approved by the KAIST Institutional Animal Care and Use Committee (KAIST IACUC). 10-week-old C57BL/6J (JAX, 000664) male mice were used for all animal experiments. Mice were maintained under a 12 h light-dark cycle in a climate-controlled (23 °C, 50% humidity), specific pathogen-free facility within the KAIST Laboratory Animal Resource Center. Standard chow diet (Envigo, 2018S) and water were provided ad libitum. Tissues were dissected and fixed for histological analysis or snap-frozen in liquid nitrogen until further analysis.

**Plasmids**. Generally, genes were amplified using PCR, digested with restriction enzymes, and ligated into plasmid vectors. C1(1–29)-TurboID-V5_pcDNA3 was a gift from Alice Ting (Addgene, #107173) and used for TurboID amplification. For the TurboID-V5-KDEL_pDisplay construct, TurboID was amplified by PCR with forward primer and a reverse primer containing V5 and KDEL. Amplicon and APEX2-V5-KDEL_pDisplay[7] vector were digested with BglII and NotI. T4 DNA ligase was used for ligation. For Sec61b-V5-TurboID_pcDNA5 construct, TurboID was amplified by PCR with forward primer containing V5 and reverse primer. The V5-TurboID amplicon was subcloned to pcDNA5 with KpnI and NotI digestion (V5-TurboID_pcDNA5). Sec61b was amplified by PCR from Sec61b-V5-APEX2_pcDNA3 (Addgene, #83411) and cloned to V5-TurboID_pcDNA5 with HindIII and KpnI digestion. Sec61b-V5-TurboID_pcDNA5 has been deposited in Addgene (#166971). For the TurboID-V5-NES_pcDNA5 construct, TurboID was amplified by PCR with forward primer and a reverse primer containing V5. TurboID-V5 amplicon was subcloned to pcDNA5 with AflII and NotI digestion (TurboID-V5_pcDNA5). NES was cloned to TurboID-V5_pcDNA with NotI and XhoI digestion. For Sec61b-V5-TurboID-APEX2_pcDNA5 construct, TurboID was amplified by PCR, and APEX2 was amplified by PCR from APEX2-V5-KDEL_pDisplay[7]. TurboID-APEX2 was amplified by overlap PCR from TurboID and APEX2 amplicons. TurboID-APEX2 and Sec61b-V5-TurboID_pcDNA5 were digested with NheI and NotI. T4 DNA ligase was used for ligation. Control pmaxGFP vector was purchased from Lonza Bioscience (Morrisville, NC, USA). Full sequences of plasmids are provided in Supplementary Data 3.

**Cell culture and transfection**. All cell lines were purchased from the American Type Culture Collection (ATCC; www.atcc.org) and cultured according to standard mammalian tissue culture protocols at 37 °C, 5% $CO_2$ in a humidified incubator. NIH-3T3 cells were cultured in DMEM (Hyclone, SH30243.01) supplemented with 10% bovine serum (Invitrogen, 16170-078) and antibiotics (100 units/mL penicillin, 100 μg/mL streptomycin). HepG2 cells were cultured in DMEM (Hyclone, SH30243.01) supplemented with 10% fetal bovine serum (Gibco, 16000-044), 1% GlutaMax (Gibco, 35050061) and antibiotics (100 units/mL penicillin, 100 μg/mL streptomycin). AML12 cells were cultured in DMEM/F12 (Gibco, 11320-033)

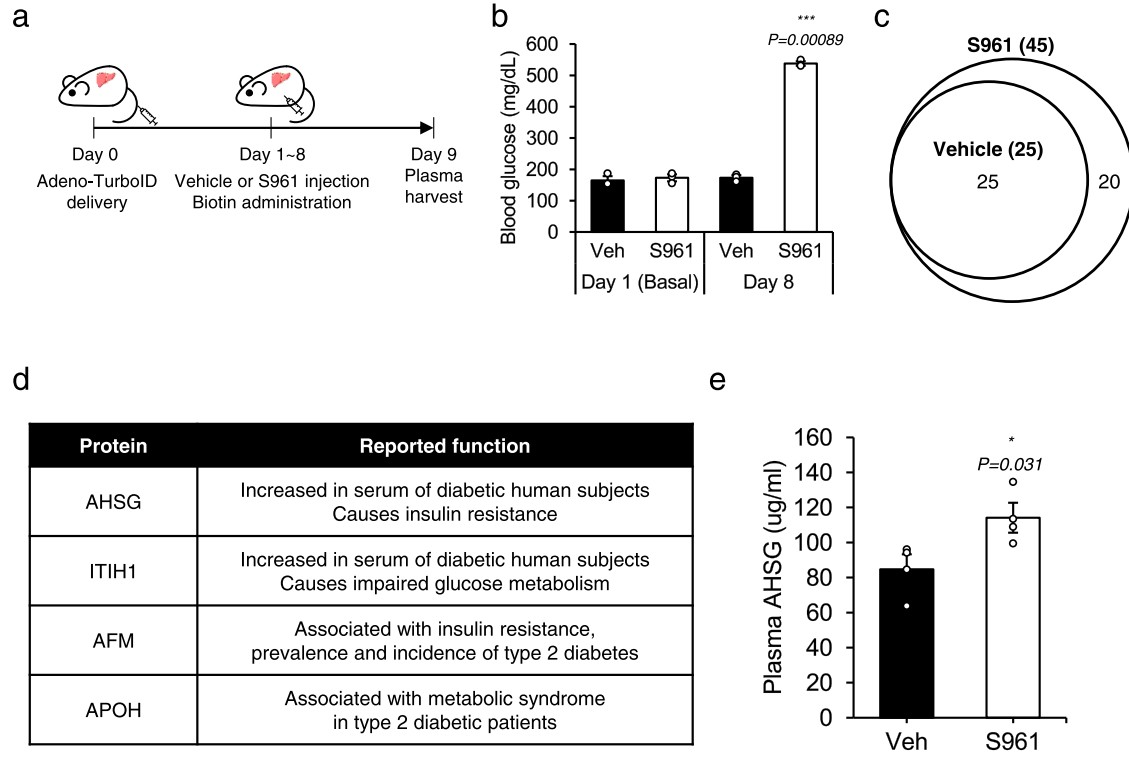

**Fig. 4 Application of iSLET to pathophysiological mouse models. a** Experimental scheme for adenoviral expression of Sec61b-V5-TurboID and biotin labeling in the S961-induced insulin resistance model. **b** Blood glucose levels in Vehicle or S961 treated mice. $n = 3$ per group. A two-tailed Student's t-test was used to compare groups. $p$-value for Day 8 group is 0.00089. **c** Venn diagram depicting biotinylated secretory proteins detected in the blood plasma of Vehicle or S961 treated liver iSLET mice from biological triplicate analysis (see Supplementary Fig. 7 for details of analysis). **d** Representative candidates related to insulin resistance detected in this study. AHSG, Alpha-2-HS-glycoprotein; ITIH1, Inter-alpha-trypsin inhibitor heavy chain H1; AFM, Afamin; APOH, Beta-2-glycoprotein 1. **e** Plasma AHSG levels of Vehicle or S961 treated mice. $n = 4$ per group. A two-tailed Student's t-test was used to compare groups. $p$-value is 0.031. All data expressed as mean ± SEM.

supplemented with 10% FBS, 1% Insulin-Transferrin-Selenium (Gibco, 41400-045) and antibiotics. 293AD cells and HeLa cells were cultured in DMEM supplemented with 10% FBS and antibiotics. Flp-In T-REx 293 cells (Invitrogen, R78007) were cultured in DMEM supplemented with 10% FBS, 2 mM L-glutamine, and antibiotics. For transient plasmid transfection, cells were plated at $2.5 \times 10^5$ cells/well in a 6-well culture plate. Twenty-four hours after plating, cells were transfected using 6 μL jetPEI (Polyplus) and 2.5 μg GFP, TurboID-KDEL, or Sec61b-TurboID plasmids according to manufacturer protocols.

**In vitro biotin labeling and cell lysate preparation**. A 5 mM biotin (Sigma, B4639) stock was prepared in DPBS with NaOH titration. Twenty-four hours after plasmid transfection or adenoviral transduction, cells were washed with PBS and further maintained for 16 h in a culture medium supplemented with 50 μM biotin. For the biotin washout experiment, following biotin labeling, cells were washed with PBS and further maintained in a fresh culture medium. Cells were lysed by RIPA (Pierce, 89901) with Xpert Protease Inhibitor Cocktail (GenDEPOT, P3100-010) and incubated for 30 min at 4 °C. Lysates were cleared by centrifugation at 16,000 × g for 20 min at 4 °C. The clear supernatant was used for western blots. Protein concentrations were determined by BCA assay (Pierce, 23225).

**Culture supernatant protein preparation for western blot**. Cells were washed with PBS twice and the culture medium was changed to phenol red-free DMEM (Hyclone, SH30284.01) supplemented 1 mM pyruvate (Sigma, S8636) with or without 50 μM biotin. For secretory pathway inhibition, 1× GolgiPlug™ (BD, 555029), which contains Brefeldin A, was treated with biotin. Sixteen hours after biotin incubation, culture supernatant was centrifuged at 400 × g for 5 min and the supernatant was filtered by 0.22 μm PES syringe filter (Millipore, SLGP033RB). The filtered supernatant was concentrated by Amicon Ultra 2 mL 10 K (Millipore, UFC201024) with buffer exchange to 50 mM Tris-HCl pH 6.8. Concentrated supernatant was used for western blots. Protein concentrations were determined by a BCA assay.

**Construction of the stably expressed Sec61b-V5-TurboID cell line**. Flp-In T-REx 293 cell lines were transfected with the pcDNA5 expression construct plasmid expressing Sec61b-V5-TurboID or TurboID-V5-NES. Cells were transfected at 60–80% confluence using 6 μL of PEI transfection reagent and 2 μg plasmid per

6-well cell culture plate. After 24 h, the cells were split into a 90 mm cell culture dish with hygromycin (2 μg/mL). Media containing hygromycin were changed every 3–4 days. After 2–3 weeks, 3–4 colonies were selected and transferred to a 24 well plate. The cells were continuously split into larger plates, and a cell stock was prepared.

**Culture supernatant protein preparation for MS analysis**. Stably expressed Sec61b-V5-TurboID or TurboID-V5-NES cell lines were cultured and each TurboID construct expression was induced by 5 ng/mL doxycycline (Sigma Aldrich). After a day, 50 μM biotin was treated for biotinylation. 16 h after biotin incubation, cells were washed with PBS twice and the culture medium was changed to phenol red-free DMEM (Hyclone, SH30284.01) supplemented 1 mM pyruvate (Sigma, S8636). After 16 h, the culture supernatant was centrifuged at 400×g for 5 min and the supernatant was filtered by 0.22 μm PES syringe filter (Millipore, SLGP033RB). The filtered supernatant was concentrated by Amicon Ultra 2 mL 10 K (Millipore, UFC201024) with buffer exchange to 50 mM Tris-HCl pH 6.8. Protein concentrations were determined by a BCA assay.

**Western blots**. Denatured proteins were separated on 12% SDS-PAGE gels. Separated proteins were transferred to the PVDF membrane (Immobilon-P, IPVH00010). Membranes were stained with Ponceau S for 15 min, washed with TBS-T (25 mM Tris, 150 mM NaCl, 0.1% Tween 20, pH 7.5) twice for 5 min, and photographed. Membranes were blocked in 3% BSA in TBS-T for 1 h, washed with TBS-T five times for 5 min each and incubated with primary antibodies, Anti-V5 (Invitrogen, R960-25, 1:10000), Anti-GAPDH (CST, 2118, 1:5000), BiP (CST, 3177, 1:1000), CHOP (CST, 2895, 1:1000) in 3% BSA in TBS-T for 16 h at 4 °C. Then, membranes were washed five times with TBS-T for 5 min each and incubated with secondary anti-mouse antibodies (Vector, PI-2000, 1:10000) or anti-rabbit antibodies (Vector, PI-1000, 1:10000) for 1 h at room temperature. For detecting biotinylated proteins, blocked membranes were incubated with streptavidin-HRP (Thermo, 21126, 1:15000) in 3% BSA in TBS-T for 1 h at room temperature. Membranes were washed five times in TBS-T before detection with chemiluminescent HRP substrate (Immobilon, P90720) and imaged on a ChemiDoc™ XRS + system (Bio-Rad, 1708265). ImageJ v.1.52v (https://imagej.nih.gov/ij/) and Image Lab v.6.0.1 (Bio-Rad) were used to quantify western blot data.

**Immunofluorescence staining**. HeLa cells were plated on round coverslips (thickness no. 1, 18 mm radius) and transfected with plasmids. Cells were treated with 50 μM biotin for 30 min. Cells were fixed with 4% paraformaldehyde and permeabilized with ice-cold methanol for 5 min at −20 °C. Next, cells were washed with DPBS and blocked for 1 h with 2% dialyzed BSA in DPBS at room temperature. Cells were incubated for 1 h at room temperature with the primary antibody, Anti-V5 (Invitrogen, R960-25, 1:5000), in a blocking solution. After washing four times with TBS-T each 5 min, cells were simultaneously incubated with secondary Alexa Fluor 488 goat anti-mouse immunoglobulin G (IgG) (Invitrogen, A-11001, 1:1000) and Streptavidin-Alexa Fluor 647 IgG (Invitrogen, S21374, 1:1000) for 30 min at room temperature. Cells were then washed four times with TBS-T each 5 min. Immunofluorescence images were obtained and analyzed using a Confocal Laser Scanning Microscope (Leica, SP8X) and Leica Application Suite X (LAS X) v.3.7.1.21655 software with White Light Laser (WLL): 470–670 nm (1 nm tunable laser) and HyD detector. For tissue imaging, livers were isolated and fixed in 10% neutral buffered formalin (Sigma, HT501128) for 24 h. Liver tissues were sectioned on a microtome at 4 μm. Tissue sections were deparaffinized in xylene, rehydrated, and antigen-retrieved. Sections were blocked in blocking solution containing 4% goat serum, 1% BSA, and 0.2% Triton X-100 in PBS for 1 h at room temperature. After blocking, sections were incubated with Streptavidin-Alexa Fluor 647 IgG (Invitrogen, S21374, 1:500) and Anti-V5 (Invitrogen, R960-25, 1:500) in blocking solution for 24 h at 4 °C. After several washes with 0.2% PBS-T, sections were incubated with Alexa Fluor 546 goat anti-Rabbit immunoglobulin G (IgG) (Invitrogen, A-11010, 1:1000) for 2 h at room temperature. Sections were washed 3 times with PBS-T, 10 min each time, mounted onto slides, and cover-slipped with a mounting solution containing DAPI (Invitrogen, P36931). Fluorescent images were captured using Zeiss LSM 780 confocal microscope. Contrast and brightness were corrected using ZEN blue 3.1 software.

**Adenovirus production and infection**. Recombinant adenoviruses were generated as previously described[27]. Briefly, Sec61b-TurboID was cloned to the pAdTrack-CMV shuttle vector by KpnI and NotI digestion. The cloned shuttle vector was linearized with PmeI and transformed to BJ5183-AD-1 cells. The recombinant adenoviral plasmid was linearized with PacI and transfected to 293AD cells. Stepwise amplification of adenovirus was performed, and adenovirus was concentrated by ViraBind™ adenovirus purification kit (Cell Biolabs, VPK-100). Adenovirus titer was measured by counting GFP-positive cells 24 h after infection with serial dilution. For adenoviral infection, cells were plated at $2.5 \times 10^5$ cells/well in a 6-well culture plate. Twenty-four hours after plating, cells were infected with $1.25 \times 10^6$ adenoviral GFP or Sec61b-TurboID particles.

**In vivo biotin labeling and protein sample preparation**. Approximately $10^8$ adenoviral GFP or Sec61b-TurboID particles were injected into mice via the tail vein. 24 mg/mL biotin stock was prepared in DMSO. Vehicle (10% DMSO in PBS) or Biotin solution (2.4 mg/mL) was filtered through a 0.22 μm PES syringe filter and injected 10 μL/g (24 mg/kg) by daily intraperitoneal injection for 3 consecutive days. For the acute insulin resistance model, S961 (100 nmol/kg, Novo Nordisk) was delivered by daily intraperitoneal injection for 8 consecutive days, 2 h prior to daily biotin injection. Biotin was not administered on the last day to minimize residual biotin in blood. Blood samples were obtained by cardiac puncture and plasma was separated in BD Microtainer® blood collection tubes (BD, 365985). Mouse Fetuin A ELISA Kit was used to determine plasma levels of fetuin A. Tissues were lysed and homogenized in RIPA buffer with Xpert Protease Inhibitor Cocktail (GenDEPOT, P3100-010) by FastPrep-24™ bead homogenizer (MP Biomedicals). Lysates were clarified by three rounds of centrifugation at $16,000 \times g$ for 20 min at 4 °C and supernatant collection. The clear supernatant was used for western blots. Protein concentrations were determined by a BCA assay.

**Peptide sample preparation and enrichment of biotinylated peptides**. Plasma and culture supernatant samples were first subjected to buffer exchange with PBS to completely remove residual free biotin via 10k MWCO filtration three times. The biotin-depleted plasma samples were transferred and denatured with 500 μL of 8 M urea in 50 mM ammonium bicarbonate for 1 h at 37 °C and followed by a reduction of disulfide bonds with 10 mM dithiothreitol for 1 h at 37 °C. The reduced thiol groups in the protein samples were subsequently alkylated with 40 mM iodoacetamide for 1 h at 37 °C in the dark. The resulting alkylated samples were diluted eight times using 50 mM ABC and subjected to trypsinization at 2% (w/w) trypsin concentration under 1 mM CaCl₂ concentration for overnight in Thermomixer (37 °C and 500 rpm). Samples were centrifuged at $10,000 \times g$ for 3 min to remove insoluble material. Then, 150 μL of streptavidin beads (Pierce, 88816) per replicate was washed with 2 M urea in TBS four times and combined with the individual digested sample. The combined samples were rotated for 1 h at room temperature. The flow-through fraction was kept, and the beads were washed twice with 2 M urea in 50 mM ABC and finally with pure water in new tubes. The bound biotinylated peptides were eluted with 400 μL of 80% acetonitrile containing 0.2% TFA and 0.1% formic acid after mixing and heating the bead slurry at 60 °C. Each eluate was collected into a new tube. The elution process was repeated four more times. Combined elution fractions were dried using Vacufuge® (Eppendorf) and reconstituted with 10 μL of 25 mM ABC for further analysis by LC-MS/MS.

**LC-MS/MS analysis of enriched biotinylated peptides**. The enriched samples were analyzed with an Orbitrap Fusion Lumos mass spectrometer (Thermo Scientific) coupled with a NanoAcquity UPLC system (Waters, Milford) in sensitive acquisition settings. Precursor ions were acquired at a range of m/z 400–1600 with 120 K resolving power and the isolation of precursor for MS/MS analysis was performed with a 1.4 Th. Higher-energy collisional dissociation (HCD) with 30% collision energy was used for sequencing with a target value of 1e5 ions determined by automatic gain control. Resolving power for acquired MS2 spectra was set to 30k at m/z 200 with 150 ms maximum injection time. The peptide samples were loaded onto the trap column (3 cm × 150 μm i.d) via the back-flushing technique and separated with a 100 cm long analytical capillary column (75 μm i.d.) packed in-house with 3 μm Jupiter C18 particles (Phenomenex, Torrance). The long analytical column was placed in a dedicated 95 cm long column heater (Analytical Sales and Services) regulated to a temperature of 45 °C. NanoAcquity UPLC system was operated at a flow rate of 300 nL/min over 2 h with a linear gradient ranging from 95% solvent A (H₂O with 0.1% formic acid) to 40% of solvent B (acetonitrile with 0.1% formic acid).

**LC-MS/MS data processing and the identification of biotinylated peptides**. All MS/MS datasets were the first subject to peak picking and mass recalibration processed with RawConverter[28] v.1.2.0.0 (http://fields.scripps.edu/rawconv) and MZRefinery[29] v.1.0 (https://omics.pnl.gov/software/mzrefinery) software, respectively, and then were searched by MS-GF+[30] algorithm v.9979 at 10 ppm precursor ion mass tolerance against the UniProt reference proteome database (55,152 entries, Mouse). The following search parameters were applied: semi-tryptic digestion, fixed carbamidomethylation on cysteine, dynamic oxidation of methionine, and dynamic biotinylation of a lysine residue (delta monoisotopic mass: +226.07759 Da). The False discovery rate (FDR) was set at <0.5% for non-redundantly biotinylated peptide level and the resulting protein FDR was near or less than 1%. MS/MS spectrum annotation for biotinylated peptides was carried out using LcMsSpectator software v.1.1 (https://omics.pnl.gov/software/lcmsspectator). Peptides with more than two spectral counts per labeled lysine site were used for the identification of bitinylated proteins. Other filtering criterion applied, if necessary, was described in the figure legend or main text.

**Histological analysis**. Mouse livers were fixed in 10% neutral buffered formalin (Sigma, HT501128) for 24 h and embedded in paraffin by an automated tissue processor (Leica, TP1020). Four micrometer-thick tissue sections were obtained, deparaffinized, rehydrated, and stained with hematoxylin and eosin.

**Statistics**. All values are expressed as the mean and standard error of the mean. A two-tailed Student's t-test was used to compare groups. P values below 0.05 were considered statistically significant. The levels of significance indicated in the graphs are $*P < 0.05$, $**P < 0.005$, and $***P < 0.001$.

**Reporting summary**. Further information on research design is available in the Nature Research Reporting Summary linked to this article.

## Data availability
Proteomics data have been deposited to the PRIDE server under accession code PXD025141. Human liver expression datasets were downloaded from the Human Protein Atlas (https://www.proteinatlas.org/). Signal peptides were predicted using the SignalP 5.0 server (http://www.cbs.dtu.dk/services/SignalP/). Gene Ontology analysis was performed on a website powered by PANTHER (http://geneontology.org/). Protein species were retrieved by UniProt Retrieve/ID mapping (https://www.uniprot.org/uploadlists/). Source data are provided with this paper. Raw images of western blots and full plasmid sequences used in this study are provided in the Source Data and Supplementary Data 3, respectively. All data that support the findings of this study are available from the corresponding author upon reasonable request. Source data are provided with this paper.

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

## Acknowledgements

We thank Dr. Kyoung-Jin Oh for assistance in adenovirus generation, Dr. Kyung Gon Kim for assistance in mass spectrometry analysis, and Hee-Saeng Jung, Ju Eun Kim, and Hannah Jung for technical support. We thank Dr. Lauge Schaffer (Novo Nordisk) for S961. We thank all lab members for helpful discussions and technical assistance. This work was supported by the National Research Foundation of Korea (NRF-2018R1A2A3075389, NRF-2016M3A9B6902871, NRF-2017K1A1A2013124, NRF-2021R1A2C200757311, NRF-2020R1C1C1013927) and the KAIST Key Research Institutes Project (Interdisciplinary Research Group). J.-S.K. was supported by the Bio and Medical Technology Development Program (NRF-2019M3E5D3073104) and the Institute for Basic Science from the Ministry of Science and ICT of Korea (IBS-R008-D1).

## Author contributions

K.K., I.P., J.-S.K., H.-W.R. and J.M.S. conceived and designed research, K.K., I.P., J.K., M.K., W.G.C. and H.S. conducted research. K.K., I.P., J.-S.K., H.-W.R. and J.M.S. analyzed data and wrote the manuscript.

## Competing interests

The authors declare no competing interests.
