## [Peer Review File · Nature Communications]

Dynamic tracking and identification of tissue-specific secretory proteins in the circulation of live miceEditorial Note: Parts of this peer review file have been redacted as indicated to maintain the confidentiality of unpublished data.

REVIEWER COMMENTS

Reviewer #1 (Remarks to the Author):

The manuscript submitted by Kim et al. with title of “Dynamic tracking and identification of tissue-specific secretory proteins in the circulation of live mice” aimed to apply the proximity labeling approach TurboID for exploring tissue-specific secretome in vivo. This is an important area to explore as secreted proteins are often critical molecules for intercellular signaling and as biomarkers. Current MS-based secretome analysis could routinely identify hundreds of secreted proteins with proper signal peptide annotation, but is limited by cytosolic protein interference. The key concept of the study is to fuse TurboID enzyme to ER-lumen anchored protein Sec61b for labeling secreted proteins which is first synthesized at ER before its transportation to the extracellular space. Proximity labeling and proteomic analysis of ER-mitochondria contact localized proteins have been recently covered in recent papers (Alice Ting et al., ACS Chem. Biol. 2019, 14, 619–635; Alice Ting et al., PNAS, 2020, 117, 12143–12154; Rhee et al., PNAS, 2020, 117, 12109–12120). The design of this study has merit for taking advantage of both secreted protein synthesis machinery and proximity labeling in living cells. However, the data for current format of the paper is too limited to approve the feasibility of the concept, especially in the in vivo setup. According to the MS-based proteomic analysis by using the advanced Orbitrap Fusion Lumos mass spectrometer, only less than 30 proteins were identified from mouse plasma sample with biotinylated peptides. This probably leads to a critical question whether the concept of this study really works efficiently and whether the in vivo proximity labeling of secreted proteins at ER could affect their proper transportation to the extracellular space. With these general concerns, significant more experimental data should be provided for approving the usefulness of this new secretome analysis approach and considering for the publication at Nature Communications. Specific comments are as following:

1. The authors provided more detailed data in cell line for demonstrating the feasibility of the whole concept. However, MS-based proteomic analysis was not performed as did in mice. Such analysis and side-by-side comparison with traditional secretome analysis is necessary to confirm the feasibility of the new approach, especially for confirming the potential negative effect of biotinylation for protein secretion. Side-by-side comparison between biotinylated cellular proteome and secretome is also necessary to confirm if this new approach could well differentiate the canonical secreted proteins, non-canonical ones and cytosolic interference.
2. By checking the western blot for culture supernatant in Fig. 1 c and e, the overall patterns look different. It's not sure whether all the data has been properly repeated with biological triplicate. Such critical information is also missing in the figure legends.

3. The key novelty and effort of this study is the in vivo application using adenoviral delivery of Sec61b-TurboID specifically to mouse liver tissue. The authors directly harvested and analyzed secreted proteins in plasma by MS. The authors properly confirmed the liver specific expression of Sec61b by western blot. However, streptavidin western blot and liver tissue slice imaging should also be performed for evaluating the biotinylation efficiency of the liver original secretome.

4. As stated above, although advanced Orbitrap Fusion Lumos mass spectrometer was applied, the authors only identified very limited types of biotinylated proteins from plasma samples. This is consistent with the limited bands in the streptavidin western blot. The identified proteins are largely the high abundant proteins which is commonly presented in plasma, and more functional lower abundant proteins are missing. This might also due to low efficient protein secretion caused by protein biotinylation? Have the authors tried to analyze the secreted proteins directly from the liver tissue sample which should get around of the high abundant biotinylated interference issue?

Reviewer #2 (Remarks to the Author):

Kim et al. describe iSLET, a method wherein proteins destined for secretion are labeled with biotin by an ER-targeted proximal biotinylation enzyme. The authors show the method in mouse liver resulting in the tracking of labeled proteins in blood plasma of the animal. iSLET enables the temporal tagging of the secretome and can thus reveal biomarkers and therapeutic targets.

The authors have indeed identified an interesting use of proximal biotinylation. Mapping the secretome in live animals will help advance this field significantly and there are many other interesting applications one can think of with this system. The data presented in the manuscript are interesting but there are some important issues to follow up.

The authors focused on liver as the source of the secretome under investigation. Since the liver is probably the most important organ contributing protein content to plasma, the MS data has to be highly reliable. Some proteins will be 'easy' to identify because of their extremely high abundance (despite very stringent search settings/FDRs) and repeat experiments will mitigate this to some extent. Therefore it is important to perform at least 3 biological replicates, with some technical replicates as well (different purifications). Supplementary table 1 shows data for two replicates but it is unclear what these replicates are (technical or biological?). It is definitely good to see that immunoglobulins are not readily detected in the analysis which supports the current data. An additional control experiment with MS on the adenoviral GFP control plasma should also be included to ensure no relevant identifications there. The need for replication is also essential for the S961 data to ensure reproducible labeling and identification. While the validation of the S961 data using literature is promising, it would be good to include some orthogonal validation using another method such as western blots or ELISA for some of the candidates.

In conclusion, the method is promising but the data presented is not sufficient to support publication at this stage.

Some minor comments:

For the in vivo experiments, mice were treated for 3 days with biotin. This would imply that labeling can last for anywhere up to almost 3 days. Labeling is also occurring on receptors, ion channels and other transmembrane proteins. This may affect the folding directly and cause the unfolded protein response in the cells. Can this provide a biased stress signature in the cells which is confounding the interpretation? Similarly, the receptors on the surface of the cells (and secreted proteins) can become inactivated because of the biotinylation. This may also affect the viability of the cells and lead to confounding factors. While short term labeling are likely less problematic, longer labeling procedures should be rigorously controlled. It is good to see that the histological analysis does not show any obvious adverse effects but it would be good to add a few markers for toxicity and ER stress to ensure no underlying issues. Note that the current study may not directly reveal these stress signatures because of the focus on liver and its important role in plasma protein production. Levels of these proteins may surpass the stress signatures but iSLET may also find applications in other tissues where the abundance of secreted proteins is likely a lot lower. Such confounding effects can be detrimental.

The following statement:

'TurboID exhibits a 100-fold improvement in efficiency compared to that of BioID in the endoplasmic reticulum (ER) of cultured human cells⁶.' suggests that TurboID is only improved in subcellular compartments. Best rephrase to make clear that the activity of TurboID is higher, independent of the subcellular compartments...

Fig1b: also show control stainings (no localization signal) + also show the overlap

Reviewer #3 (Remarks to the Author):

I find there is not enough detail about the constructs the authors are using to be able to judge the manuscript. It is not clear how their KDEL-Turbo and Sec61-Turbo constructs are created, and in particular what the topology of the Sec61b-Turbo construct is going to be, to be able to critically judge the data. They do cite a previous paper of theirs that describes similar constructs with an APEX2 tag, but these are entirely new constructs, and in the previous paper they described several arrangements of Sec61b-APEX2, but it is not clear what sort of arrangement with Turbo they've

created here. Given that Sec61b is a tail-anchored protein, the possibilities for disrupted or heterogenous topology confounding the results are substantial, and it is not productive to try and evaluate the data without knowing how the constructs are organized and, importantly, having evidence that they do in fact uniformly achieve the correct topology (showing mere ER localization is insufficient for this purpose). For this reason, the manuscript in its present form is unsuitable.

Reviewer #1 (Remarks to the Author):

Reviewer's comments:

The manuscript submitted by Kim et al. with title of "Dynamic tracking and identification of tissue-specific secretory proteins in the circulation of live mice" aimed to apply the proximity labeling approach TurboID for exploring tissue-specific secretome in vivo. This is an important area to explore as secreted proteins are often critical molecule for intercellular signaling and as biomarker. Current MS-based secretome analysis could routinely identifies hundreds of secreted proteins with proper signal peptide annotation, but is limited by cytosolic protein interference. The key concept of the study is to fuse TurboID enzyme to ER lumen anchored protein Sec61b for labeling secreted proteins which is firstly synthesized at ER before its transportation to the extracellular space. Proximity labeling and proteomic analysis of ER-mitochondria contact localized proteins have been recently covered in recent papers (Alice Ting et al., ACS Chem. Biol. 2019, 14, 619–635; Alice Ting et al., PNAS, 2020, 117, 12143-12154; Rhee et al., PNAS, 2020, 117, 12109-12120). The design of this study has its merit for taking advantage of both secreted protein synthesis machinery and proximity labeling in living cells. However, the data for current format of the paper is too limited to approve the feasibility of the concept, especially in the in vivo setup.

According to the MS-based proteomic analysis by using the advanced Orbitrap Fusion Lumos mass spectrometer, only less than 30 proteins were identified from mouse plasma sample with biotinylated peptides. This probably leads to a critical question whether the concept of this study really work efficiently and whether the in vivo proximity labeling of secreted proteins at ER could affect their proper transportation to the extracellular space. With these general concerns, significant more experimental data should be provided for approving the usefulness of this new secretome analysis approach and the considering for the publication at Nature Communications. Specific comments are as following:

Response:

We thank the reviewer for providing an accurate and detailed evaluation of our Sec61b-TurboID technology which, drawing upon our understanding of secreted protein synthesis machinery and proximity labeling enzymes, provides a robust methodology to analyze organ-specific secretomes.

The reviewer also raises valid concerns regarding the number of proteins identified from mass spectrometry and Sec61b-TurboID labeling efficiency of secretory proteins. However, we view the seemingly modest number of proteins identified not as a problem of labeling efficiency but rather as a reflection of the exceptional selectivity of Sec61b-TurboID labeling of ER-transit proteins and high-fidelity Spot-BioID mass spectrometry workflow which is optimized for the exclusive detection of biotinylated peptides. Of mouse plasma proteins, 70% is liver-derived albumin and ~100s proteins are routinely detected in mouse plasma without albumin

depletion. As shown in **Supplementary Table 2**, the number of proteins we detected after streptavidin enrichment was 160~180 for both control and experimental groups, which is well within reason.

We performed further analysis with Spot-BioID, a method we previously developed to exclude false positives, i.e., non-biotinylated proteins. Briefly, Spot-BioID is an analysis workflow that allows detection of biotinylated residues from each mass spectrometry result via mass shift and, through this process, we were able to dramatically reduce, if not effectively eliminate, false positives such as non-specific binding proteins (Lee, S.Y. et al. *ACS Cent. Sci.* **2**, 506-516 (2016) and Lee, S.Y. et al. *J. Am. Chem. Soc.* **139**, 3651-3662 (2017)).

In Adeno-GFP control mouse plasma, 160~170 proteins were detected in the primary mass data, but among them, we did not detect any biotinylated peptides which is consistent with the absence of any bands in the streptavidin-HRP blot of control plasma (**Supplementary Table 2** and **Fig. 3c**). In Adeno-Sec61b-TurboID plasma, 40~50 biotinylated proteins were clearly identified, and 27 proteins with ≥ 3 of biotinylated spectral counts were finally obtained. We believe this number is within reasonable range considering the stringency imposed by the high selectivity of Sec61b-TurboID labeling combined with exceptional fidelity of Spot-BioID analysis.

Streptavidin enriched sample name	Replicate #	Before filtering out non-biotinylated spectra		After filtering out non-biotinylated spectra (Spot-BioID)			
		Total peptide spectral counts	Total proteins	Biotinylated peptide spectral counts	Biotinylated proteins	Reproducibly identified biotinylated proteins	Proteins with ≥ 3 of biotinylated spectral counts
Adeno-GFP + Biotin	1	1055	163	0	0	0	0
	2	1355	174	0	0		
Adeno-Sec61b-TurboID + Biotin	1	2480	189	418	45	42	27
	2	2644	185	538	56		

(Supplementary Table 2 in the revised Supplementary Information)

(Fig. 3c and 3d in the revised manuscript)

Reviewer's comments:

1. The authors provided more detailed data in cell line for demonstrating the feasibility of the whole concept. However, MS-based proteomic analysis was not performed as did in mice. Such analysis and side-by-side comparison with traditional secretome analysis is necessary to confirm the feasibility of the new approach, especially for confirming the potential negative effect of biotinylation for protein secretion. Side-by-side comparison between biotinylated cellular proteome and secretome is also necessary to confirm if this new approach could well differentiate the canonical secreted proteins, non-canonical ones and cytosolic interference.

Response:

We agree with the suggestion to provide additional mass data to evaluate the feasibility of our concept in cell line studies. Towards this end, we have now compared cytosolic labeling (TurboID-NES) and ER lumen labeling (Sec61b-TurboID) to confirm that our ER lumen labeling approach effectively differentiates canonical secreted proteins from cytoplasmic proteins. We performed proteomic analysis of biotinylated proteins enriched from culture supernatant of HEK293 cells expressing either TurboID-NES or Sec61b-TurboID.

We detected 184 and 504 proteins in the culture supernatant of TurboID-NES expressing cells and Sec61b-TurboID expressing cells, respectively. Gene Ontology (GO) analysis showed that cytosol and cytoplasmic proteins were top ranked in TurboID-NES expressing cells (**Fig. 2c**) whereas extracellular proteins were top ranked in Sec61b-TurboID expressing cells (**Fig. 2d**). We also analyzed the annotated subcellular localizations of detected proteins in each group and found, as expected, the fraction of secreted proteins to be highly enriched in Sec61b-TurboID expressing cells (**Fig. 2e**). These results provide additional evidence for Sec61b-TurboID function in cell line studies.

(Fig. 2 in the revised manuscript)

Reviewer's comments:

2. By checking the western blot for culture supernatant in Fig. 1 c and e, the overall patterns look different. It's not sure whether all the data has been properly repeated with biological triplicate. Such critical information is also missing in the figure legends.

Response:

The reviewer correctly points out that the western blots in Fig. 1c and 1e looks different. This is due to secretome differences in cell lines as Fig. 1c is a blot for proteins secreted from NIH-3T3 cells and 1e is a blot for proteins secreted from HepG2 cells. The secretome patterns from each cell line was highly reproducible and experiments were performed in biological triplicates. We now add this information to the figure legend.

Reviewer's comments:

3. The key novelty and effort of this study is the *in vivo* application using adenoviral delivery of Sec61b-TurboID specifically to mouse liver tissue. The authors directly harvested and analyzed secreted proteins in plasma by MS. The authors properly confirmed the liver specific expression of Sec61b by western blot. However, streptavidin western blot and liver tissue slice imaging should also be performed for evaluating the biotinylation efficiency of the liver original secretome.

Response:

Per reviewer suggestions, we performed immunofluorescence imaging on liver tissue sections to evaluate the efficiency of Sec61b-TurboID labeling activity at the cellular level. We confirmed that TurboID (Anti-V5) was expressed in the liver of mice transduced with Sec61b-TurboID adenovirus, and that proteins were efficiently biotinylated (stained by Streptavidin-AlexaFluor after biotin administration (**Fig. 3b**).

(Fig. 3b in the revised manuscript)

Reviewer's comments:

4. As stated above, although advanced Orbitrap Fusion Lumos mass spectrometer was applied, the authors only identified very limited types of biotinylated proteins from plasma samples. This is consistent with the limited bands in the streptavidin western blot. The identified proteins are largely the high abundant proteins which is commonly presented in plasma, and more functional lower abundant proteins are missing. This might also due to low efficient protein secretion caused by protein biotinylation? Have the authors tried to analyze the secreted

proteins directly from the liver tissue sample which should get around of the high abundant biotinylated interference issue?

Response:

We have addressed the reviewer's concern regarding the number of labeled plasma proteins by liver Sec61b-turboID above. Regarding the reviewer's suggestion to analyze *ex vivo* secretomes, the main goal of the study was to detect secretory proteins of the liver *in situ* in live mice, so we did not perform *ex vivo* liver explant experiments. However, we surely recognize that *ex vivo* applications of Sec61b-turboID will also hold value under experimental conditions that preclude the use of live organisms. For example, we have analyzed human liver organoid secretomes using Sec61b-TurboID and find that Sec61b-TurboID secretome analysis offers tremendous advantages over conventional methods that directly analyze culture supernatant proteomes.

Reviewer #2 (Remarks to the Author):

Reviewer's comments:

Kim et al. describe iSLET, a method wherein proteins destined for secretion are labeled with biotin by an ER-targeted proximal biotinylation enzyme. The authors show the method in mouse liver resulting in the tracking of labeled proteins in blood plasma of the animal. iSLET enables the temporal tagging of the secretome and can thus reveal biomarkers and therapeutic targets. The authors have indeed identified an interesting use of proximal biotinylation. Mapping the secretome in live animals will help advance this field significantly and there are many other interesting applications one can think of with this system. The data presented in the manuscript are interesting but there are some important issues to follow up.

The authors focused on liver as the source of the secretome under investigation. Since the liver is probably the most important organ contributing protein content to plasma, the MS data has to be highly reliable. Some proteins will be 'easy' to identify because of their extremely high abundancy (despite very stringent search settings/FDRs) and repeat experiments will mitigate this to some extent. Therefore it is important to perform at least 3 biological replicates, with some technical replicates as well (different purifications). Supplementary table 1 shows data for two replicates but it is unclear what these replicates are (technical or biological?). It is definitely good to see that immunoglobulins are not readily detected in the analysis which supports the current data. An additional control experiment with MS on the adenoviral GFP control plasma should also be included to ensure no relevant identifications there. The need for replication is also essential for the S961 data to ensure reproducible labeling and identification.

Response:

To maximize the accuracy and fidelity of Sec61b-TurboID labeled secretory protein analysis, we performed mass spectrometry data analysis using Spot-BioID, a method we previously developed to exclude false positives, i.e., non-biotinylated proteins, from analysis. Briefly, Spot-BioID is an analysis workflow that allows detection of biotinylated residues from each mass spectrometry result via mass shift. Through this process, we dramatically reduce, if not effectively eliminate, false positives such as non-specific binding proteins (Lee, S.Y. et al. *ACS Cent. Sci.* **2**, 506-516 (2016) and Lee, S.Y. et al. *J. Am. Chem. Soc.* **139**, 3651-3662 (2017)).

As the reviewer points out, we have already performed mass spectrometry analysis from technical duplicates. In the Adeno-GFP control mouse plasma, 160~170 proteins were detected in the primary mass data, but among them, we did not detect any biotinylated peptides which is consistent with the absence of any bands in the streptavidin-HRP blot of control plasma (**Supplementary Table 2** and **Fig. 3c**). In the Adeno-Sec61b-TurboID mouse plasma, 40~50 biotinylated proteins were clearly identified, and 27 proteins with ≥ 3 biotinylated spectral counts were obtained. We are confident that this is a highly accurate and reliable list of proteins that satisfy stringent criteria imposed by the selectivity of Sec61b-TurboID labeling combined with high fidelity identification of true positives by Spot-BioID analysis.

Streptavidin enriched sample name	Replicate #	Before filtering out non-biotinylated spectra		After filtering out non-biotinylated spectra (Spot-BioID)			
		Total peptide spectral counts	Total proteins	Biotinylated peptide spectral counts	Biotinylated proteins	Reproducibly identified biotinylated proteins	Proteins with ≥ 3 of biotinylated spectral counts
Adeno-GFP + Biotin	1	1055	163	0	0	0	0
	2	1355	174	0	0		
Adeno-Sec61b-TurboID + Biotin	1	2480	189	418	45	42	27
	2	2644	185	538	56		

(Supplementary Table 2 in the revised Supplementary Information)

(Fig. 3c and 3d in the revised manuscript)

Nevertheless, we fully agree with the reviewer's suggestion that increasing the number of mass spectrometry samples will enhance our current data. To this end, we repeated mass spectrometry analysis for vehicle and S961 groups in biological triplicates per group. As a result, our new data set shows 25 biotinylated proteins in the vehicle treated group plasma and 45 proteins in the S961 treated group plasma (**Supplementary Figure 7**).

(Supplementary Figure 7a and 7b in the revised Supplementary Information)

Reviewer's comments:

While the validation of the S961 data using literature is promising, it would be good to include some orthogonal validation using another method such as western blots or ELISA for some of the candidates. In conclusion, the method is promising but the data presented is not sufficient to support publication at this stage.

Response:

For orthogonal validation, in an independent experiment, we quantified the plasma concentration of AHSG (Fetuin-A) with a commercially available ELISA assay and confirmed that the plasma level of AHSG was increased by ~35% in the mice treated with S961 relative to controls (**Fig. 4e**).

(Fig. 4e in the revised manuscript)

Reviewer's comments:

Some minor comments: For the in vivo experiments, mice were treated for 3 days with biotin. This would imply that labeling can last for anywhere up to almost 3 days. Labeling is also occurring on receptors, ion channels and other transmembrane proteins. This may affect the folding directly and cause the unfolded protein response in the cells. Can this provide a biased stress signature in the cells which is confounding the interpretation? Similarly, the receptors on the surface of the cells (and secreted proteins) can become inactivated because of the biotinylation. This may also affect the viability of the cells and lead to confounding factors. While short term labeling are likely less problematic, longer labeling procedures should be rigorously controlled. It is good to see that the histological analysis does not show any obvious adverse effects but it would be good to add a few markers for toxicity and ER stress to ensure no underlying issues.

Response:

We agree that the biotinylation of ER-transit proteins by Sec61b-TurboID may have spurious effects perturbing normal mouse physiology. To assess this possibility, we checked expression levels of ER stress markers in liver tissue of mice transduced with Sec61b-TurboID adenovirus and treated with biotin. We found that levels of the well-described ER stress markers BiP and CHOP in liver tissue were not affected by Sec61b-TurboID labeling activity. Furthermore, liver sections

immunostained for cleaved caspase-3, an apoptosis marker, was also unaltered by Sec61b-TurboID labeling activity. These data show that Sec61b-TurboID labeling activity, under our current experimental conditions, does not have any discernable effects on normal liver physiology, including ER-stress and apoptosis (**Supplementary Fig. 5**).

(Supplementary Fig. 5 in the revised Supplementary Information)

Reviewer's comments:

Note that the current study may not directly reveal these stress signatures because of the focus on liver and its important role in plasma protein production. Levels of these proteins may surpass the stress signatures but iSLET may also find applications in other tissues where the abundance of secreted proteins is likely a lot lower. Such confounding effects can be detrimental.

Response:

While we did not detect any changes in ER stress or apoptosis in our liver studies, we agree that this aspect should be carefully examined in a case-by-case manner as the iSLET application expands to other tissues. Furthermore, we agree

that the cumulative effects of sustained *i*SLET labeling activity, as opposed to transient expression by adenovirus transduction coupled to short-term biotin supplementation, on normal mouse physiology should be carefully examined in future applications of *i*SLET technology.

Reviewer's comments:

The following statement: 'TurboID exhibits a 100-fold improvement in efficiency compared to that of BioID in the endoplasmic reticulum (ER) of cultured human cells6.' suggests that TurboID is only improved in subcellular compartments. Best rephrase to make clear that the activity of TurboID is higher, independent of the subcellular compartments... Fig1b: also show control stainings (no localization signal) + also show the overlap

Response:

We agree with the reviewer's comment regarding the clarifications on TurboID activity and have made appropriate changes in the manuscript. We also add cellular localization images of TurboID-NES as a comparative control for cytosol-localized TurboID in **Fig. 1b**.

(Fig. 1b in the revised manuscript)

Reviewer #3 (Remarks to the Author):

Reviewer's comments:

I find there is not enough detail about the constructs the authors are using to be able to judge the manuscript. It is not clear how their KDEL-Turbo and Sec61-

Turbo constructs are created, and in particular what the topology of the Sec61b-Turbo construct is going to be, to be able to critically judge the data.

Response:

We apologize for omitting important information regarding construct structure and thank the reviewer for pointing out this mistake. KDEL is a four amino acid peptide sequence that acts as an ER retention signal when fused to the C-terminus of a target protein via recognition by the ER-resident KDEL receptor. The successful application of TurboID-KDEL (C-terminus fusion) for ER lumen targeting has been described elsewhere (Branon, T.C. et al. *Nat. Biotechnol.* **36**, 880-887 (2018)). On the other hand, Sec61b is a single-pass ER transmembrane protein of well-defined topology with its N and C terminus exposed to the cytosol and ER lumen, respectively (Mavylutov, T. et al. *Protein Cell* **9**, 733–737 (2018)). We fused TurboID to the C-terminus of Sec61b (luminal side) to target the ER lumen. The revised manuscript now contains detailed cloning procedures along with construct information to clarify these points.

Reviewer`s comments:

They do cite a previous paper of theirs that describes similar constructs with an APEX2 tag, but these are entirely new constructs, and in the previous paper they described several arrangements of Sec61b-APEX2, but it is not clear what sort of arrangement with Turbo they've created here. Given that Sec61b is a tail-anchored protein, the possibilities for disrupted or heterogenous topology confounding the results are substantial, and it is not productive to try and evaluate the data without knowing how the constructs are organized and, importantly, having evidence that they do in fact uniformly achieve the correct topology (showing mere ER localization is insufficient for this purpose). For this reason, the manuscript in its present form is unsuitable.

Response:

The Sec61b-TurboID fusion protein expression construct was created by an in-frame insertion of TurboID to the C-terminus of Sec61b (Sec61b-TurboID) and we have revised the manuscript and figures (**Supplementary Fig. 1a**) to describe this point more clearly. Amino acid sequences and full DNA sequences of plasmids used in this study are also provided in **Supplementary Table 3**. To experimentally verify the proposed topology of the Sec61b-TurboID fusion protein, we performed peroxidase-assisted EM imaging. We fused APEX2 to the C-terminus of Sec61b-TurboID (N-Sec61b-TurboID-APEX2-C), and confirmed that the TurboID-APEX2 domain is indeed exposed to the ER lumen based upon peroxidase staining patterns (**Supplementary Fig. 1b**) which show typical ER lumen staining patterns (Martell, J.D. et al. *Nat. biotechnol.*, **30**, 1143-1148 (2012) and Mavylutov, T. et al. *Protein Cell*

a

b

(Supplementary Fig. 1 in the revised Supplementary Information)

In addition, we compared cytosolic labeling (TurboID-NES) and ER lumen labeling (Sec61b-TurboID) to further demonstrate specific labeling of ER lumen proteins by Sec61b-TurboID. To this end, we performed proteomic analysis of biotinylated proteins enriched from culture supernatant of TurboID expressed HEK293 cells via liquid chromatography and tandem mass spectrometry (LC-MS/MS).

We detected 184 and 504 proteins in the culture supernatant of TurboID-NES expressing cells and Sec61b-TurboID expressing cells, respectively. Gene Ontology (GO) analysis showed that cytosol and cytoplasmic proteins were top ranked in TurboID-NES expressing cells (**Fig. 2c**) whereas extracellular proteins were top ranked in Sec61b-TurboID expressing cells (**Fig. 2d**). We also analyzed the annotated subcellular localizations of detected proteins in each group and found, as

expected, the fraction of secreted proteins to be highly enriched in Sec61b-TurboID expressing cells (**Fig. 2e**). These results provide further evidence that Sec61b-TurboID is correctly localized to the ER lumen.

(Fig. 2 in the revised manuscript)

REVIEWER COMMENTS

Reviewer #1 (Remarks to the Author):

We noticed that there is a Nat. Chem. Biol. Paper (Cell type-selective secretome profiling in vivo, pubmed id. 33199915) published end of last year on almost the same topic which significantly diluted the novelty of current study. Although the ER-targeted strategy is the same, the depth and quality of current study is far from the published one. Current manuscript should therefore be significantly improved and extended in a complimentary manner. Otherwise, the qualification to be published on Nat. Commun. is questionable.

1. For response to comment 1, the authors failed to address my comments directly. To clarify, the traditional secretome analysis approach is done by collecting conditioned media from cultured cells and analyzing by MS. In addition, according to the Nat. Chem. Biol. paper, proximity targeting cytosolic protein could also label secreted proteins. This is not consistent with the manuscript's conclusion.

2. For response to comment 4, the authors mentioned "we have analyzed human liver organoid secretomes using Sec61b-TurboID and find that Sec61b-TurboID secretome analysis offers tremendous advantages over conventional methods that directly analyze culture supernatant proteomes." Such type of data is critical to strength the current manuscript, especially with the presence of published study on the same topic.

3. The authors should also analyze the non-biotinylated peptides as did in the Nat. Chem. Biol. paper to increase their coverage for protein identification. Current dataset is too limited.

Reviewer #2 (Remarks to the Author):

The authors have adequately addressed my comments.

Reviewer #3 (Remarks to the Author):

I had one substantive comment on the previous version of the paper, which is that the constructs were not adequately described, which prevented me from thoroughly analyzing the results. Fortunately, the authors have rectified that by describing more clearly the constructs that were made and providing evidence that the Sec61b-TurboID construct is likely to be topologically correct. This revision allowed me to evaluate the manuscript more fully.

This is a methods paper, in which the authors describe a system with potential *in vivo* utility for labeling the secretome. Although this paper is itself a fairly modest proof-of-concept, the potential applications are fairly obvious, and might supplant more cumbersome methods like ¹³C or D₂O labeling, that are not specific for just secretory pathway clients. I find the data to be of reasonable quality and the conclusions to be warranted.

The degree to which the system is useful depends on (a) how uniformly secretory client proteins are labeled and (b) that the system not grossly perturb ER function. I think the authors provide data that get to both points, but in my opinion there are a few simple experiments that would more clearly address these criteria:

1. To point (a), how uniform is biotinylation of proteins in the ER lumen relative to the ER luminal content? If the ER were purified from either cells or liver, and then the luminal contents were released by low concentrations of some non-ionic detergent and separated from the membranes, would the biotinylated proteins from that sample, assessed by SA-HRP, look very similar in composition to the total content, assessed by Coomassie Stain or similar?
2. Also to point (a), and somewhat related to #1 above, does the TurboID label only nascent proteins as they translocate, as one might expect given its fusion to Sec61b, or can it essentially label anything within the ER lumen, including resident proteins? The simplest way to address this would be to ask in cells to what extent inhibiting protein synthesis with cycloheximide prevents labeling by TurboID.
3. To point (b), I am unconvinced by the ER stress data presented in Figure S5. For one thing, from Fig. S8, the BiP appears to migrate a bit too big (BiP is usually very close to the 75kDa marker) and CHOP too small (CHOP is usually about halfway between 25 and 37). I grant that different gels migrate things a little bit differently. What I'd really like to see is a positive control—for example a tunicamycin-injected liver, 1mg/kg for 12-16h—showing robust upregulation of BiP and CHOP.

Reviewer #1 (Remarks to the Author):

Reviewer's comments:

We noticed that there is a Nat. Chem. Biol. Paper (Cell type-selective secretome profiling in vivo, pubmed id. 33199915) published end of last year on almost the same topic which significantly diluted the novelty of current study. Although the ER-targeted strategy is the same, the depth and quality of current study is far from the published one. Current manuscript should therefore be significantly improved and extended in a complimentary manner. Otherwise, the qualification to be published on Nat. Commun. is questionable.

1-1. For response to comment 1, the authors failed to address my comments directly. To clarify, the traditional secretome analysis approach is done by collecting conditioned media from cultured cells and analyzing by MS.

Response:

(Reviewer only figure)

As the reviewer requests, we now performed traditional secretome analysis and compared the results with cytosolic labeling (TurboID-NES) or ER luminal labeling (Sec61b-TurboID). We detected 301 proteins in the traditional secretome analysis (a),

184 and 504 biotinylated proteins in culture supernatant of TurboID-NES expressing cells and Sec61b-TurboID expressing cells, respectively (b and c). Traditional secretome analysis produced significant overlap with biotinylated proteome data from cytoplasmic or ER luminal labeling (d). We further analyzed the annotated subcellular localization of detected proteins from each group and found that the fraction of cytoplasmic proteins and secreted proteins was 29% and 28% in traditional secretome analysis. As expected, cytoplasmic proteins were highly enriched (52%) in culture supernatant of TurboID-NES expressing cells in contrast to 14% in culture supernatant of Sec61b-TurboID expressing cells (e). We added a summary of these new results in Supplementary Figure 4 to support the conclusion that Sec61b-TurboID labels canonical secretory proteins with similar coverage and specificity as the traditional secretome analysis method in cultured cells.

1-2. In addition, according to the Nat. Chem. Biol. paper, proximity targeting cytosolic protein could also label secreted proteins. This is not consistent with the manuscript's conclusion.

Response:

[Redacted]

There seems to be some misunderstanding about the difference between “classical” secreted proteins and “unconventionally” secreted proteins. As stated in the *Nat. Chem. Biol.* Paper, “Each proximity labeling construct was cotransfected with a vector encoding a **classically secreted protein** (peptidase M20 domain containing 1 (**PM20D1**) or an **unconventionally secreted protein** (**FGF1**). (...) Secreted PM20D1 was robustly biotinylated by the two lumenally oriented constructs Mem-TurboID and ER-TurboID but not by Cyto-TurboID (Fig.1b). (...) Secreted FGF1 was robustly biotinylated by Cyto-TurboID and to a lesser extent by the two other lumenally oriented constructs”. That is, PM20D1, a “classically” secreted protein, is not labeled by Cyto-TurboID, and only FGF1, an “unconventionally” secreted protein, is labeled by Cyto-TurboID. Both the *Nat. Chem. Biol.* paper and our study reach the same conclusion that ER-TurboID (Sec61b-TurboID) labels “classically” secretory proteins.

Reviewer's comments:

2. For response to comment 4, the authors mentioned “we have analyzed human liver organoid secretomes using Sec61b-TurboID and find that Sec61b-TurboID secretome analysis offers tremendous advantages over conventional methods that directly analyze culture supernatant proteomes.” Such type of data is critical to strength the current manuscript, especially with the presence of published study on the same topic.

Response:

(Reviewer only figure)

A major advantage we found during the organoid studies is that biotin-labeled secretory proteins can be readily detected even in the presence of full culture media which is required to maintain healthy organoid cultures. We provide data regarding

this point in the streptavidin-HRP blot of Sec61b-TurboID labeled secretomes from human liver organoid culture supernatant. Although there are many proteins in culture supernatant that are not synthesized and secreted by the liver organoids, e.g. bovine serum albumin from organoid culture media (indicated by arrow), Sec61b-TurboID labeled secretory proteins were easily enriched from culture supernatant by streptavidin bead pulldown and could be detected on a streptavidin-HRP blot (see lanes labeled as E). While Sec61b-TurboID offers an advantage over traditional secretome analysis in this aspect, the organoid application is still an *in vitro* system and therefore adds little conceptual value to the main conclusion and novelty of the current manuscript which demonstrates that Sec61b-TurboID can label tissue-specific secreted proteins that can be tracked and detected in the blood circulation of live mice.

Reviewer's comments:

3. The authors should also analyze the non-biotinylated peptides as did in the *Nat. Chem. Biol.* paper to increase their coverage for protein identification. Current dataset is too limited.

Response:

Scheme 1. MS Analysis of Biotin-Labeled Proteins Produced by Proximity Labeling

Spot-BioID workflow (Lee, S.Y. et al. ACS Cent Sci 2, 506-516 (2016))

(Reviewer only figure)

In Supplemental Table 1 of the *Nat. Chem. Biol.* paper, the authors list 60 statistically significant liver-derived secretory proteins detected in plasma. They used AAV-Tbg-ER-TurboID virus for labeling of classical secretory proteins of liver. When we compared their data with our data, they failed to detect the well-known liver-derived secretory proteins Alb, Hpx, and Kng1, whereas they detected apparent false positive proteins such as Fstl5, Tns4, and Ca2, which are not expressed in liver and also do not contain signal sequences. Based on these results, we do not see any advantage or new insight to be gained by analyzing the non-biotinylated fraction of proteins to simply increase coverage. Therefore, we feel that our current results utilizing the Spot-BioID workflow (Lee et al, 2016) adequately supports the main conclusion that the biotinylated plasma proteome Sec61b-TurboID efficiently detects the *in vivo* liver secretome.

Reviewer #2 (Remarks to the Author):

Reviewer's comments:

The authors have adequately addressed my comments.

Reviewer #3 (Remarks to the Author):

Reviewer's comments:

I had one substantive comment on the previous version of the paper, which is that

the constructs were not adequately described, which prevented me from thoroughly analyzing the results. Fortunately, the authors have rectified that by describing more clearly the constructs that were made and providing evidence that the Sec61b-TurboID construct is likely to be topologically correct. This revision allowed me to evaluate the manuscript more fully.

This is a methods paper, in which the authors describe a system with potential in vivo utility for labeling the secretome. Although this paper is itself a fairly modest proof-of-concept, the potential applications are fairly obvious, and might supplant more cumbersome methods like 13C or D2O labeling, that are not specific for just secretory pathway clients. I find the data to be of reasonable quality and the conclusions to be warranted.

The degree to which the system is useful depends on (a) how uniformly secretory client proteins are labeled and (b) that the system not grossly perturb ER function. I think the authors provide data that get to both points, but in my opinion there are a few simple experiments that would more clearly address these criteria:

1. To point (a), how uniform is biotinylation of proteins in the ER lumen relative to the ER luminal content? If the ER were purified from either cells or liver, and then the luminal contents were released by low concentrations of some non-ionic detergent and separated from the membranes, would the biotinylated proteins from that sample, assessed by SA-HRP, look very similar in composition to the total content, assessed by Coomassie Stain or similar?

Response:

(Supplementary Figure 2)

There are technical limitations to isolate ER protein with high purity due to cross-contamination with other organelles that are in tight contact with the ER. Instead, we used two strategies to label ER proteins in this study: TurboID-KDEL, which is

uniformly distributed within the ER lumen, and Sec61b-TurboID, which is tethered to the luminal face of the ER membrane. In Fig.S2, we show that the patterns of the labeled proteins in cell lysates of TurboID-KDEL and Sec61b-TurboID are nearly identical. This indicates that Sec61b-TurboID, though tethered to the ER membrane, can label the same population of ER lumen proteins that are labeled by TurboID-KDEL. The high labeling efficiency of ER lumen proteins by Sec61b-TurboID is likely due to Sec61b being a component of the ER translocon complex allowing Sec61b-TurboID to label both non-membrane ER luminal proteins and ER membrane proteins alike.

Reviewer's comments:

2. Also to point (a), and somewhat related to #1 above, does the TurboID label only nascent proteins as they translocate, as one might expect given its fusion to Sec61b, or can it essentially label anything within the ER lumen, including resident proteins? The simplest way to address this would be to ask in cells to what extent inhibiting protein synthesis with cycloheximide prevents labeling by TurboID.

Response:

(Supplementary Figure 1 in the revised manuscript)

We thank the reviewer for raising this point which we have also been investigating. We performed the cycloheximide (CHX) experiment to evaluate the labeling properties of Sec61b-TurboID towards nascently synthesized vs post-synthesis ER-resident proteins. Our results show that a significant fraction of ER-resident proteins was labeled in cycloheximide-treated cell lysates, however, in the case of culture supernatant, our results indicate that labeled secretory proteins were mostly of newly synthesized proteins. We added these new results in Supplementary Figure 1.

Reviewer's comments:

3. To point (b), I am unconvinced by the ER stress data presented in Figure S5. For one thing, from Fig. S8, the BiP appears to migrate a bit too big (BiP is usually very close to the 75kDa marker) and CHOP too small (CHOP is usually about halfway between 25 and 37). I grant that different gels migrate things a little bit differently. What I'd really like to see is a positive control—for example a tunicamycin-injected liver, 1mg/kg for 12-16h—showing robust upregulation of BiP and CHOP.

Response:

(Supplementary Figure 6 and uncropped images in the revised manuscript)

Again, we thank the reviewer for careful assessment of our data. We repeated the experiment using liver lysates from tunicamycin injected mice (1mg/kg, 16 h) as a positive control and evaluated expression of BiP and CHOP. The protein size was exactly as the reviewer points out and we now present data with the appropriate bands indicated for BiP and CHOP. However, there was still no difference in the expression levels of BiP and CHOP in Sec61b-TurboID expressing liver compared with liver of vehicle injected mouse indicating that Sec61b-TurboID expression and labeling has insignificant effects on ER stress in liver tissue.

REVIEWERS' COMMENTS

Reviewer #3 (Remarks to the Author):

My critiques have been addressed adequately.